# Sympatho-Vagal Dysfunction in Systemic Sclerosis: A Follow-Up Study

**DOI:** 10.3390/life13010034

**Published:** 2022-12-23

**Authors:** Gabriel Dias Rodrigues, Angelica Carandina, Costanza Scatà, Chiara Bellocchi, Lorenzo Beretta, Pedro Paulo da Silva Soares, Eleonora Tobaldini, Nicola Montano

**Affiliations:** 1Department of Clinical Sciences and Community Health, University of Milan, 20122 Milan, Italy; 2Post-Graduation Program in Cardiovascular Sciences, Fluminense Federal University, Niterói 24020-141, Brazil; 3Department of Internal Medicine, Fondazione IRCC.S. Ca’ Granda, Ospedale Maggiore Policlinico, 20122 Milan, Italy; 4Department of Physiology and Pharmacology, Fluminense Federal University, Niterói 24020-141, Brazil

**Keywords:** heart rate variability, scleroderma, autoimmune diseases, autonomic nervous system

## Abstract

Systemic sclerosis (SSc) patients often present cardiovascular autonomic dysfunction, which is associated with the risk of arrhythmic complications and mortality. However, little is known regarding the progression of cardiac autonomic impairment over time. We aimed to evaluate the cardiac autonomic modulation among SSc with limited cutaneous (lcSSc), diffuse cutaneous (dcSSc) subset, and age-matched healthy control (HC) at baseline (t0) and five-year follow-up (t1). In this follow-up study, ECG was recorded at t0 and t1 in twenty-four SSc patients (dcSSc; *n* = 11 and lcSSc; *n* = 13) and 11 HC. The heart rate variability (HRV) analysis was conducted. The spectral analysis identified two oscillatory components, low frequency (LF) and high frequency (HF), and the sympatho-vagal balance was assessed by the LF/HF ratio. The LF/HF increased (*p* = 0.03), and HF reduced at t1 compared to t0 in dcSSc (*p* = 0.03), which did not occur in the lcSSc and HC groups. Otherwise, both lcSSc and dcSSc groups presented augmented LF/HF at t0 and t1 compared to HC (*p* < 0.01). In conclusion, a worsening of cardiac autonomic dysfunction is related to the dcSSc subset, in which a more extent of skin fibrosis and internal organs fibrosis is present.

## 1. Introduction

Systemic Sclerosis (SSc) is a complex immune-mediated disease distinguished by vascular impairment, autoantibodies production, and fibrosis. SSc is a rare systemic autoimmune disease with an incidence of 0.6–2.3:100.000 in Europe [1] and a standardized mortality ratio of 3.5 [2]. The first sign of SSc is the Raynaud phenomenon (RP), an uncontrolled vasospasm of the small vessels. Several specific SSc autoantibodies are produced, and irreversible organ fibrosis, as a result of chronic ischemia, affects skin and internal organs, such as heart, lungs, the gastrointestinal tract with any organ that potentially can be affected [3,4]. Heart and lungs disease involvement causes several clinical manifestations and represents the major cause of mortality among SSc patients [5]. Based on the extent of their fibrosis, SSc is classified into two subsets: the limited (lcSSc) and diffuse cutaneous (dcSSc) subsets [6]. SSc patients have a higher mortality risk when compared to other rheumatic diseases, in particular the dcSSc subset [7]. Cardiac involvement is associated with up to 70% mortality at 5 years [8,9] and may occur independently from other typical complications of SSc [10].

Heart rate variability (HRV) has been described as a powerful non-invasive tool to access heart sympathetic and vagal modulations [11]. The reduction of vagal-mediated HRV indexes was also associated with the risk of arrhythmic complications and mortality in SSc patients [12,13]. Also, autonomic dysfunction is an early marker of SSc progression and could precede cardiac fibrosis occurrence helping to identify subclinical cardiac involvement. Overall, SSc patients have a predominant sympathetic modulation and vagal withdrawal compared to age-matched healthy controls [14,15].

From a clinical point of view, chronic sympatho-vagal imbalance is a potent risk factor for dangerous cardiovascular events and mortality [16]. The sympathetic overactivity is associated with an increased cardiovascular workload, endothelial dysfunction, coronary spasm, left ventricular hypertrophy, and serious arrhythmias [17]. On the other side, the augmented vagal activity exerts a protective effect against ischemia and decreases heart rate and blood pressure [18].

Cardiac autonomic impairment was also associated with high values of pulmonary arterial pressure (PAPs) in SSc patients without a diagnosis of pulmonary hypertension (PH), unmasking a preclinical derangement of the cardiopulmonary circulation in SSc [19]. In our previous cohort study, both dcSSc and lcSSc subsets presented impaired cardiac autonomic control when compared to a healthy control group. Moreover, dcSSc showed an augmented sympathetic and reduced vagal cardiac modulation if compared to lcSSc patients [15]. Lastly, symptoms of dysautonomia (e.g., esophageal dysmotility, diarrhea, and occlusive syndrome) reduce SSc patients’ quality of life [20,21]. Namely, impairments in cardiac autonomic control are associated with low quality of life scores and with sleep disturbances in SSc [22].

Although the aforementioned studies support the clinical relevance of cardiac autonomic monitoring in SSc, the progression of a sympatho-vagal imbalance over time has not been studied yet. Thus, we conducted a five-year follow-up study to investigate the progression of cardiac autonomic dysfunction in lcSSc and dcSSc patients in comparison to a healthy control group.

## 2. Materials and Methods

### 2.1. Sample

This study included 24 SSc patients (dcSSc; *n* = 11 and lcSSc; *n* = 13), and 11 healthy control (HC) subjects matched by age and sex. SSc patients were enrolled from the Scleroderma Unit (Immunology and Allergology Department, Fondazione IRCC.S. Ca’ Granda, Ospedale Maggiore Policlinico, Milan, Italy), while HC subjects were enrolled from the community.

For the SSc group, we applied the study’s inclusion criteria enrolling dcSSc or lcSSc subsets based on the extent of their skin fibrosis [23]; patients with a definite SSc but without skin fibrosis yet with puffy fingers were categorized in the lcSSc group. We considered exclusion criteria: (1) diabetes mellitus; (2) coronary arterial disease; (3) the absence of a stable sinus rhythm on ECG; and (4) ongoing therapy with beta-blocker. For the HC group, the use of any medications with cardiovascular or metabolic action or the presence of cardiovascular or metabolic diseases were applied as exclusion criteria.

Clinical and laboratory parameters within 3 months from study visits were extracted from medical records. Forced vital capacity (FVC), diffusing capacity for carbon monoxide (DLCO), and SSc-associated autoantibodies (anti-centromere antibodies—ACA, anti-nuclear antibodies—ANA, and anti-Scl-70 positivity) were collected from routine tests. PH were diagnosed according to a standard recommendation [24].

The protocol was approved by the local Ethics Committee (Comitato Etico Milano Area 2, 682_2017 approved at 9 November 2017), and all participants signed informed written consent before participation in the study.

### 2.2. Experimental Design

For each patient, two standard experimental sessions were performed at different time points: baseline (t0) and after five years for the follow-up assessment (t1).

SSc patients and HC subjects underwent the recording of ECG and respiration by an ad hoc telemetric system device equipped with a thoracic piezoelectric belt (LAB3, Marazza Spa, Monza, Italia). The signals were recorded at rest, in supine position for 10 min with spontaneous breathing. Systolic (SBP) and diastolic (DBP) blood pressures were obtained by the auscultatory method using a standard sphygmomanometer (GIMA Spa, Milano, Italia). Mean blood pressure (MBP = DBP +1/3 × [SBP—DBP]) was calculated. All blood pressure records were conducted by a single experienced operator.

### 2.3. Heart Rate Variability Analysis

The heart rate variability (HRV) was evaluated through specific software (Heart Scope II, AMPS, Italia). A single selected segment of 300 beats without artifacts was selected from each ECG recording. Spectral analysis was performed through the autoregressive model, with a Hanning window and 50% overlap to obtain the spectral power in the low frequency (LF, frequency band bounded between 0.04 and 0.15 Hz) and high frequency (HF, frequency band bounded between 0.15 and 0.40 Hz and synchronous with respiration) components. The LF and HF components were expressed in both absolute value and normalized units (LFnu and HFnu), obtained by dividing each band power by the total power subtracted from the very low component (<0.04 Hz). The autonomic balance was calculated as the LFnu/HFnu ratio [11].

Non-linear dynamics of HRV were evaluated by symbolic analysis. The R-R dynamics were classified into three families: (a) patterns with no variation (0V; all three symbols were equal); (b) patterns with one variation (1V; two consequent symbols were equal and the remaining symbol was different); and (c) patterns with two liked (2LV) and unlike (2UV) variations (e.g., all symbols were different from the previous one). The percentage of the patterns with no variation (0V) and with two variations (2UV or 2LV) are marker of sympathetic and parasympathetic cardiac autonomic modulation, respectively. However, the pattern of 1V was not associated with any autonomic tests used to validate the method [25].

### 2.4. Statistics

The sample size was calculated (statistical power >0.80) from the main outcome variables (LF and HF). The current sample provided a statistical power of 0.81. To calculate statistical power, we inputted the following parameters: α< 0.05; number of groups (3), number of measurements (2) and effect size (directly calculated by partial η^2^) [26].

The Shapiro-Wilk test was employed to evaluate the normality of data distribution. Two-way ANOVA was used for a within-between analysis (factors time and groups) with a repeated measurement in the first factor (time) and a Holm-Sidak’s multiple comparisons post-hoc test. One-way ANOVA for independent measurements was used to compare age among dcSSc, lcSSc, and HC groups. Chi-squared test was used to compare other demographic and clinical variables.

The *p*-value <0.05 was considered statistically significant. Data were presented as mean ± SD. The software used was SPSS Statistics version 21.0 (IBM Corp., Armonk, NY, USA) and GraphPad Prism version 8.0 (GraphPad Software Inc., San Diego, CA, USA).

## 3. Results

Although thirty-seven SSc patients (*n* = 18 dcSSc and *n* = 19 lcSSc) and nineteen HC were enrolled in the first assessment of the current study, thirteen SSc (*n* = 7 dcSSc and *n* = 6 lcSSc) and seven HC dropped out. The dropout reasons were reported in Figure 1. We excluded four patients (*n* = 3 dcSSc and *n* = 1 lcSSc) from the final analysis at t1 due to the development of arrhythmias that compromised the HRV analysis. The data of four patients (*n* = 1 dcSSc and *n* = 3 lcSSc) were missed since patients lost follow-up at our center. Participants who dropped by medical reasons (e.g., stroke, hemodialysis, lung transplantation, cancer, and orthopedic surgery) declined to participate in the follow-up (t1) assessment. The flowchart of the current study is presented in Figure 1.

Table 1 shows demographics and clinical features at follow-up assessment (t1). As expected, dcSSc presented a reduced FVC and DLCO compared to lcSSc. and ACA positivity was observed only in the lcSSc group. As regards current medications, mostly of patients were under calcium channel blockers and low dose aspirin for the control of Raynaud’s phenomenon (RP). Immunosuppressant treatments were conducted in 64% of dcSSc and 38% of lcSSc patients. No differences were found between groups across the time points (t0 vs. t1) regarding demographics and clinical features.

Table 2 summarizes the hemodynamic data among lcSSc, dcSSc, and HC subjects at t0 and t1. HR was higher in SSc patients compared to HC at t0 and t1, with no changes across time and within the SSc subsets (lcSSc vs. dcSSc). The DBP and MBP were higher in dcSSc than lcSSc, with no interaction with time.

Both lcSSc and dcSSc showed an augmented sympathetic-mediated and reduced vagal-mediated HRV indexes, and a shift of the sympatho-vagal balance towards a sympathetic predominance (increased LF/HF) compared to HC at baseline. After five-years follow-up (t1), dcSSc patients showed a more pronounced sympathetic predominance (increased LF n.u and LF/HF) and vagal withdrawal (reduced HF n.u) compared to baseline data (t0). In contrast, lcSSc patients and HC subjects did not show changes in HRV indexes over time. No differences were found in the HRV total power index among groups at t0 (HC 1061 ± 1371; lSSc 1091 ± 1882; dSSc 886 ± 477 ms^2^; *p* = 0.97) and t1 (HC 768 ± 812; lSSc 638 ± 716; dSSc 736 ± 438 ms^2^; *p* = 0.85). These results are presented in Figure 2 and Figure 3 (inter-individual variability from t0 to t1).

## 4. Discussion

The main findings from the current study were the following: (1) SSc patients (both lcSSc and dcSSc) presented a shifted sympatho-vagal towards sympathetic predominance and vagal withdrawal when compared to HCs at baseline and over time (after five years’ follow-up); and (2) the sympatho-vagal dysfunction worsened over time only in SSc patients with dcSSc subset.

To our knowledge, cardiovascular autonomic dysfunction in SSc patients was first documented in a classical study that employed a simple approach to HR responses to isometric hand-grip exercise. It was reported that SSc patients had blunted tachycardia to exercise compared to healthy controls [27]. Also, SSc patients showed an attenuated HR response to vagal-mediated autonomic tests (e.g., Valsalva maneuver and deep breathing test), suggesting an impairment of the cardiac neural control through the parasympathetic branch from the autonomic nervous system [27].

Thenceforward, the cardiovascular autonomic control in SSc has been assessed by different approaches, including the HRV analysis. The HRV is a non-invasive tool to access cardiac sympathetic and vagal modulation [11]. In SSc disease, sympathetic overactivity and reduced vagal modulation are reported by recent studies [13,15,19,22,27]. Also, a reduction of vagal-mediated HRV is a predictor of e myocardial damage and ventricular arrhythmias [12,13] and is associated with low quality of life and sleep scores in SSc patients [22].

Studies that investigated HRV profile stratifying for SSc subsets (i.e., lcSSc and dcSSc) showed conflicting results. A study did not find any differences between SSc subsets [28], while two studies found a reduced HRV total power in dcSSc compared to lcSSc [14,15]. The reduced HRV total power should be compatible with a cardiac autonomic dysfunction from both branches of the autonomic nervous system as observed in other rheumatologic [29,30] and cardiopulmonary [31,32] diseases.

The imbalance between the sympathetic and parasympathetic branches of the ANS is a hallmark of Immune-Mediated Inflammatory Diseases, influencing the inflammatory processes, innate and adaptive immunity [33]. From experimental studies, microinfusions of pro-inflammatory cytokines (e.g., TNF-α, IL-6, and IL1β) in cardiovascular nuclei (hypothalamus and medulla oblongata) increased the sympathetic and reduced parasympathetic activity [34]. In human studies, an augmented cardiac sympathetic modulation was found in the early phase of rheumatoid arthritis and SSc, suggesting that autonomic dysfunction precedes or predicts the onset of the clinical disease stage [35]. Thus, the link between ANS and inflammation seems to be bidirectional, but the question of who comes first remains unanswered.

In the current study, we observed that cardiac autonomic control dysfunction progressed over time only in dcSSc patients. The sympathetic predominance increased while the vagal-mediated HRV was reduced after five years. An explanation could be that dcSSc are mainly characterized by extended skin fibrosis as well as a higher internal organs fibrosis (especially interstitial lung disease) [23]. We can suppose that fibrosis may provoke a deteriorative effect on the cardiac structures, offsetting the sympatho-vagal to an augmented sympathetic modulation and vagal withdrawal that follows the disease progression. Indeed, some putative mechanisms have been considered to explain cardiovascular autonomic dysfunction in SSc disease. SSc may provoke a deteriorative effect in microcirculatory systems, impairing pulmonary and cardiac structures [6]. It reduces the integrity and the number of receptors and exerts a general inhibitory influence on cardiovascular control centers [36]. Therefore, afferent information could be interrupted and offset the sympatho-vagal balance to sympathetic predominance and vagal withdrawal at rest overactivity at rest [6,36] in SSc patients.

Recently, chronic inflammation plays a role in the extracellular matrix deposition process leading to cardiac fibrosis [37,38], and chronic low-grade inflammation is related to augmented cardiac sympathetic modulation and vagal withdrawal in SSc [39]. Thus, the chronic inflammation and sympathetic overactivity might be connected in a bidirectional way, suggesting potential targets for therapies in SSc disease.

To the best of our knowledge, this is the first study investigating the cardiovascular autonomic control in SSc patients after a follow-up. Thus, the sympatho-vagal offset in SSc patients with diffuse fibrosis (dSSc) over time suggests a worsening of autonomic dysfunction associated with the disease’s severity. A strength of our study is that we followed-up a healthy control group. No differences were found in HRV indexes between baseline and follow-up in HC, suggesting no effect of natural aging on cardiac autonomic control. On the other hand, comparisons among SSc and HC groups confirmed that autonomic dysfunction were kept stable after five years in SSc, despite the subset (lcSSc and dcSSc).

From our results, both SSc subsets (lcSSc and dSSc) are tachycardic compared to the healthy control group at baseline and after the follow-up (Table 2). On the other hand, systolic and diastolic blood pressures were not different between SSc subsets and HC at baseline or follow-up. The dSSc group had a slight increase in diastolic and mean blood pressures at follow-up assessment compare to lcSSc group. This increase seems to be not clinically relevant and we did not have further data to support possible underlying mechanisms. However, a recent study suggested that SSc patients have a reduced arterial baroreflex sensitivity compared to healthy subjects [29], which should be considered for future follow-up protocols.

Moreover, our study suggests that HRV indexes are relevant markers to be included in the clinical screening and monitoring of SSc since it is a non-invasive and accessible tool. Cardiac autonomic control could be a potential therapeutic target in SSc and for future studies pharmacological and non-pharmacological approaches that stimulate the vagus nerve and reduce sympathetic activity should be considered to counteract the autonomic dysfunction of SSc [40,41].

## 5. Conclusions

The cardiac autonomic dysfunction worsens over time in SSc with a more severe phenotype of the disease, the dcSSc. The development of augmented cardiac sympathetic and reduced vagal modulation is compatible with an advanced fibrosis status. However, the study has limitations. The small sample size is a main limitation. The assessment of cardiac autonomic modulation of preclinical SSc subjects (very earlySSc) that our group assessed at baseline in a previous study, could have been of great interest to follow-up.

## Figures and Tables

**Figure 1 life-13-00034-f001:**
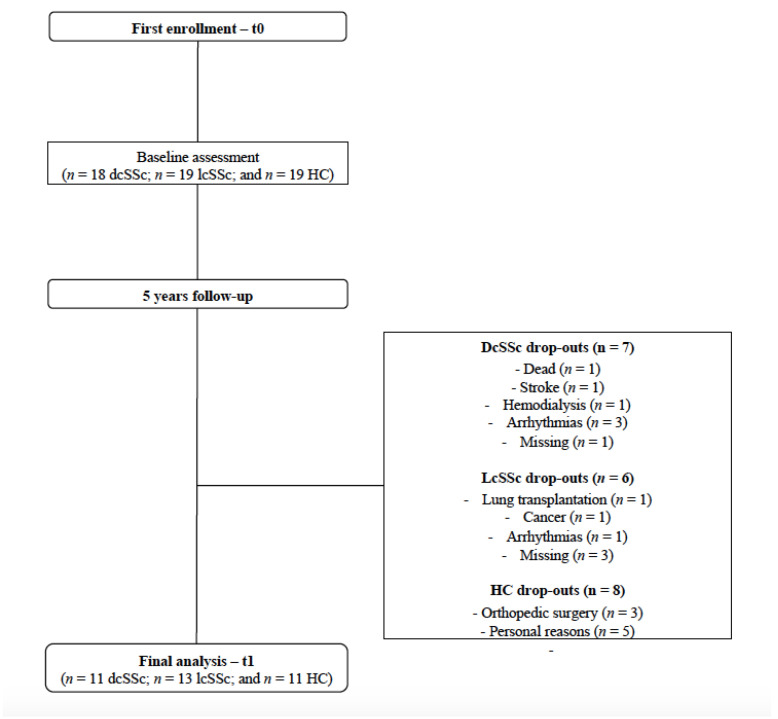
The study’s flowchart.

**Figure 2 life-13-00034-f002:**
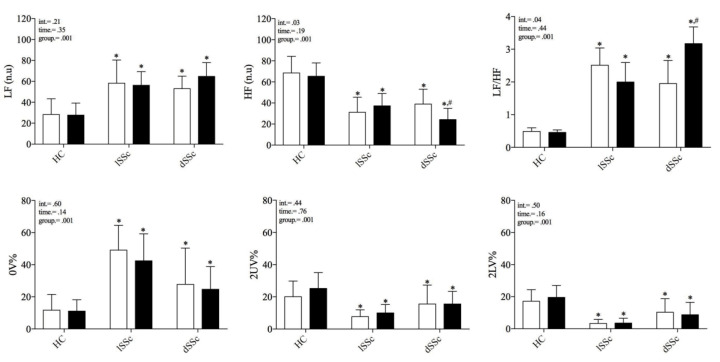
Comparison of cardiac autonomic modulation at rest between systemic sclerosis subsets and healthy control individuals before and after 5-years follow-up. SSc: systemic sclerosis; lcSSc: limited cutaneous SSc (*n* = 11); dcSSc: diffuse cutaneous (*n* = 8); SSc; HC: age-matched healthy control group (*n* = 11); HF (n.u.): high frequency normalized unity; LF (n.u.): low frequency normalized unity; 0V%: patterns with no variations; 2LV%: patterns with two like variations; 2ULV%: patterns with two unlike variations. Empty and black bars represent t0 and t1, respectively. * differences from HC group; # differences from baseline (t0). differences accounted by Holm-Sidak’s post-hoc.

**Figure 3 life-13-00034-f003:**
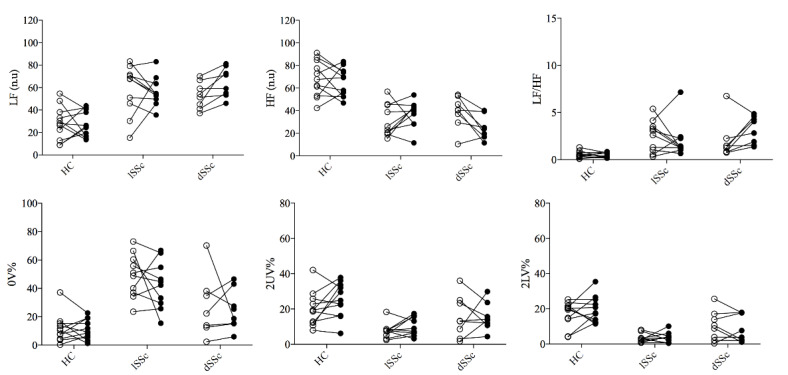
Inter-individual variability of cardiac autonomic control before and after 5-years follow-up. SSc: systemic sclerosis; lcSSc: limited cutaneous SSc (*n* = 11); dcSSc: diffuse cutaneous (*n* = 8); SSc; HC: age-matched healthy control group (*n* = 11); HF (n.u.): high frequency normalized unity; LF (n.u.): low frequency normalized unity; 0V%: patterns with no variations; 2LV%: patterns with two like variations; 2ULV%: patterns with two unlike variations. Empty and black dots represent t0 and t1, respectively.

**Table 1 life-13-00034-t001:** Clinical characteristics of SSc patients and healthy controls.

	HC	lcSSc	dcSSc	*p*-Value
Demographic and clinical data				
N	11	13	11	-
Age (years)	63 ± 6	61 ± 8	60 ± 10	0.40
Gender (F/%)	9 (82)	12 (92)	7 (64)	0.21
Etnicity Caucasian (*n*/%)	11 (100)	13 (100)	11 (100)	-
Disease duration (years)	-	18 ± 8	23 ± 9	0.26
PH (*n*/%)	-	4 (31)	3 (27)	0.49
HPN (*n*/%)	-	8 (62)	6 (46)	0.61
Antibodies				
ANA+ (*n*/%)	-	10 (76.9)	10 (90.9)	0.60
ACA+ (*n*/%)	-	7 (53.8)	0 (0)	0.01
Anti-Scl-70+ (*n*/%)	-	2 (15.4)	6 (54.5)	0.08
Spirometry				
FVC (%)	-	105 ± 18	83 ± 15	0.02
DLCO (%)	-	78 ± 17	52 ± 18	0.01
Echocardiography				
LVEF (%)	-	62 ± 6	61 ± 5	0.48
TPSE (mm)	-	21 ± 4	22 ± 3	0.84
PAPs (mmHg)	-	31 ± 10	32 ± 11	0.61
Medications				
Diuretics (*n*/%)	-	5 (38)	3 (27)	0.30
ACEi or ARB (*n*/%)	-	3 (23)	4 (36)	0.59
CC.B. (*n*/%)	-	5 (38)	5 (45)	0.23
Immunosuppressants (*n*/%)	-	3 (23)	7 (64)	0.32
ERAs (*n*/%)	-	4 (31)	3 (27)	0.49
Low dose aspirin (*n*/%)	-	7 (54)	7 (63)	0.61

SSc: systemic sclerosis; lcSSc: limited cutaneous SSc; dcSSc: diffuse cutaneous; HC: age-matched healthy control group. PH: pulmonary hypertension; ANA+: anti-nuclear antibodies positive; ACA +: Anti-centromere antibodies positive; Anti-Scl-70+: Anti-topoisomerase I antibodies positive; FVC: forced vital capacity; DLCO: Diffusing capacity for carbon monoxide; ERAs: Endothelin receptor antagonists; ACEi: Angiotensin-converting enzyme inhibitors; ARBs: Angiotensin receptor blockers; CC.B.s: Calcium channel blockers; HPN: arterial hypertension. One-way ANOVA for independent measurements was used to compare age among dcSSc, lcSSc and HC groups; unpaired *t* test was employed to compare FVC and DLCO; chi-squared test was used to compare other variables.

**Table 2 life-13-00034-t002:** Comparison of hemodynamic variables between systemic sclerosis subsets and healthy control individuals before and after 5-years follow-up.

	HC	lcSSc	dcSSc	*p*-Value		
	(*n* = 11)	(*n* = 13)	(*n* = 11)	Int.	Time	Group
HR (bpm)						
t0	66 ± 10	79 ± 13 *	76 ± 10 *	0.86	0.49	0.02
t1	67 ± 10	79 ± 9 *	76 ± 9 *			
SBP (mmHg)						
t0	122 ± 12	120 ± 8	125 ± 8	0.10	0.15	0.43
t1	123 ± 9	130 ± 16	125 ± 5			
DBP (mmHg)						
t0	76 ± 7	76 ± 8	78 ± 4	0.41	0.59	0.02
t1	75 ± 7	71 ± 8	82 ± 6 ^$^			
MBP (mmHg)						
t0	92 ± 8	91 ± 7	94 ± 5	0.83	0.68	0.03
t1	91 ± 7	91 ± 9	96 ± 5 ^$^			

SSc: systemic sclerosis; lcSSc: limited cutaneous SSc; dcSSc: diffuse cutaneous; HC: age-matched healthy control group. Two-way ANOVA for independent measurements was used to comparisons between time and groups factors; among dcSSc, lcSSc and HC groups. int.: interaction; * differences HC group; $ differences from LcSSc; differences accounted by Holm-Sidak’s post-hoc.

## Data Availability

The datasets analyzed are available from the corresponding author upon reasonable request.

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
