# Peer review of "Sympatho-Vagal Dysfunction in Systemic Sclerosis: A Follow-Up Study"

_life, 2022, doi:10.3390/life13010034_

Round 1

Reviewer 1 Report

The study entitled "Sympatho-vagal dysfunction in systemic sclerosis: a follow-up study" is a good attempt by the authors to investigate the dysfunction of sympathetic and vagal systems. However, I suggest several major modifications before considering further. My comments are as follows;

Abstract

1. The background information on scleroderma must be provided here with the important issues associated with SS.

2. An outline on the methodology must be provided in the abstract

Introduction

3. Authors must detail the significance and impact of sympatho-vagal dysfunction as a separate paragraph

4. A more clear information on systemic sclerosis must be included in the introduction, especially to make aware about the global statistics of the disease and mechanism of autoimmune stimulation

Methods

5. Whether the authors followed any inclusion and exclusion criteria; it must be stated in the methods.

6. Why the authors limitted their parameters to the heart rate and pressure variations

Discussion

7. The discussion should contain more information of sympatho-vagal dysfunction in various diseases and also emphasize the possible reason for the same. 

Author Response

Reviewer 1

The study entitled "Sympatho-vagal dysfunction in systemic sclerosis: a follow-up study" is a good attempt by the authors to investigate the dysfunction of sympathetic and vagal systems. However, I suggest several major modifications before considering further. My comments are as follows;

Abstract

  1. The background information on scleroderma must be provided here with the important issues associated with SS.
  2. An outline on the methodology must be provided in the abstract

Reply. We appreciate the reviewer’s suggestion. We included new information regarding the background and methodology. Although the strict format (max 200 words), we hope these further sentences hope abstract readability. 

Introduction

  1. Authors must detail the significance and impact of sympatho-vagal dysfunction as a separate paragraph

Reply. We thank the reviewer for this relevant comment. We included a new paragraph highlighting the clinical significance and impact of sympatho-vagal dysfunction.  Please see page 2 lines 48-53.

  1. A more clear information on systemic sclerosis must be included in the introduction, especially to make aware about the global statistics of the disease and mechanism of autoimmune stimulation

Reply. We thank the reviewer, and add information on disease incidence, mortality and pathogenesis in the introduction. Please see page 1 lines 28-34

Methods

  1. Whether the authors followed any inclusion and exclusion criteria; it must be stated in the methods.

Reply. Thank you for this observation. We added a new paragraph detailing the inclusion and exclusion criteria. Please see page 2 lines 74-82.

  1. Why the authors limited their parameters to the heart rate and pressure variations

Reply. We thank the opportunity to clarify this point. The main goal of the current study is to follow up on the cardiovascular autonomic dysfunction over time in SSc subsets compared to healthy controls. As a secondary outcome, we evaluated clinical parameters, such as spirometry and echocardiographic data (now included), which did not change over time.

Discussion

  1. The discussion should contain more information of sympatho-vagal dysfunction in various diseases and also emphasize the possible reason for the same. 

Reply. The reviewer is correct. In the revised manuscript, we included a paragraph discussing the sympatho-vagal dysfunction and its possible reasons that may explain this phenomenon in rheumatologic diseases. Please see pages 7-8 lines 270-279.

Reviewer 2 Report

Manuscript ID: life-1974657

 Sympatho-vagal dysfunction in systemic sclerosis: a follow-up study

Autonomic fucntions may be also influenced by comborbidities

Autonomic fuctnionsmay be also determined by the current medcation

10/22

Author Response

Reviewer 2

Sympatho-vagal dysfunction in systemic sclerosis: a follow-up study

Autonomic functions may be also influenced by comorbidities

Reply. We thank the reviewer for this concern. Comorbidities with a direct effect on ANS such as diabetes and coronary arterial disease were applied as exclusion criteria. We reformulated the table 1, and included new information about comorbidities. Although comorbidities can influence the autonomic function, no differences were found between SSc subsets and across t0 and t1 periods. So, the groups are equivalent regarding the prevalence of potential comorbidities that may influence the cardiovascular autonomic control, such as arterial hypertension and pulmonary hypertension.

Autonomic functions may be also determined by the current medication

Reply. We thank the reviewer for this comment that is correct. Medications can influence the heart rate variability and the autonomic balance. In this specific case, as per exclusion criteria, patients were not under beta blockers (that are the main drug able to influence the autonomic functions) and in general SSc patients do not tolerate betablockers that can cause and worsen the Raynaud Phenomenon (RP). Calcium channel blockers and low dose aspirin are the medications prescribed to SSc patients for the control of RP and, to the best our knowledge, do not have direct influence on the ANS. Same observation for immunosuppressants that were an ongoing treatment in 65% dcSSc and in 38% lcSSc.  We added a sentence in the results section, to specify the therapies. Please see page 4 lines 162-168.             

Reviewer 3 Report

In this manuscript, the authors attempt to show longitudinal change of sympatho-vagal balance in systemic sclerosis (SSc). Although the point of view is relatively novel and interesting, there are several serious flaws in their methodology.  

Major comments:

1.     2.4. Statistics: Two-way ANOVA test is inappropriate for this setting because data on t0 and t1 are paired.

2.     3. Results: The authors should explain the reason why patients were ruled out from the study due to stroke, hemodialysis, lung transplantation, cancer, and orthopedic surgery.

3.     Table 1: Description of demographics is too poor. The information should be displayed for both t0 and t1. Moreover, some important features of SSc, such as disease duration and skin thickness score, are lacking. How about echocardiography findings?

4.     Figure 2: Each data should be displayed in before-after dot plots to illustrate longitudinal change in each subject.

Minor comments:

1.     2.4. Statistics: From which time point data of LF and HF did the authors calculate the sample size?

2.     2.4. Statistics: Post-hoc analysis should be conducted not by Tukey’s test but by Tukey-Kramer’s test because the number of subjects differs among dcSSc, lcSSc, and HC groups.

3.     Table 2: I suppose “HR (bpm)” should be written one line down.

Author Response

Reviewer 3

In this manuscript, the authors attempt to show longitudinal change of sympatho-vagal balance in systemic sclerosis (SSc). Although the point of view is relatively novel and interesting, there are several serious flaws in their methodology.  

Major comments:

2.4. Statistics: Two-way ANOVA test is inappropriate for this setting because data on t0 and t1 are paired.

Reply. We thank the reviewer for the opportunity to clarify this point. We used the two-way ANOVA for a within-between analysis (time and groups factors). So, the first factor (time) was considered a repeated measurement. We included a sentence in the statistics section to make it clear for the readers. Please see page 3 lines 130-131.

  1. Results: The authors should explain the reason why patients were ruled out from the study due to stroke, hemodialysis, lung transplantation, cancer, and orthopedic surgery.

Reply. We appreciated the opportunity to clarify it. We contacted these patients for follow-up assessment, but they refused participation for these reasons. We included a new sentence in the results section to clarify it. Please see page 3 lines 144-155.  

Table 1: Description of demographics is too poor. The information should be displayed for both t0 and t1. Moreover, some important features of SSc, such as disease duration and skin thickness score, are lacking. How about echocardiography findings?

Reply. We thank the reviewer for this concern. We added in table 1 the echocardiography data, disease duration and other clinical features in the table 1. No differences were found between SSc subtypes and across the time points (t0 vs. t1). Unfortunately, we have no complete data for all patients about the skin thickness score (mRSS) especially for lcSSc that clinicians not always report in the medical charts. We confirm that all patients had skin involvement with extension according to their specific subset.

 Figure 2: Each data should be displayed in before-after dot plots to illustrate longitudinal change in each subject.

Reply. We appreciated this suggestion. We included new graphs with before-after dot plots (Figure 3). We believe that will improve the interpretation from the readers about the inter-individual variability.

Minor comments:

2.4. Statistics: From which time point data of LF and HF did the authors calculate the sample size?

Reply. We thank the opportunity to clarify this point. The sample size was calculated considering an average statistical power ≥0.80 for the main outcome variables LF and HF. The current sample enrolled provided a statistical power of 0.81. To calculate sample power, we inputted the following parameters: α< 0.05; number of groups (3), number of measurements (2) and effect size (directly calculated by partial η2).  We include new sentences in the statistics section. Please see page 3 lines 124-126.

2.4. Statistics: Post-hoc analysis should be conducted not by Tukey’s test but by Tukey-Kramer’s test because the number of subjects differs among dcSSc, lcSSc, and HC groups.

Reply. We thank the reviewer for this concern, and apologize for this mistake. Indeed, we conducted the "Holm-Sidak's multiple comparisons test", which is also strong recommended for the number of subjects differs between groups.    

Table 2: I suppose “HR (bpm)” should be written one line down.

Reply. Yes, it is correct now.   

Round 2

Reviewer 1 Report

Authors must include a description on the limitations of the study along with the conclusion.

Author Response

We appreciate the reviewer’s suggestion. The description of Study's limitations was included along with the conclusion.

Reviewer 3 Report

The authors appropriately responded to all my comments.

Author Response

We thank the reviewer for his/her relevant comments that helped a lot to improve the manuscript.